# Living with and managing type 1 diabetes in humanitarian settings: A qualitative synthesis of lived experience and stakeholder tacit knowledge

Oria James[1], Linda Abbou-Abbas[2,3], Lavanya Vijayasingham[4]*

**1** MSc Public Health Graduate Class of 2023, London School of Hygiene & Tropical Medicine, London, United Kingdom, **2** International Committee of the Red Cross, Beirut Delegation, Lebanon, **3** INSPECT-LB (Institut de Santé Publique, Epidemiologie Clinique et Toxicologie-Liban), Beirut, Lebanon, **4** NCD in Humanitarian Settings Research Group and Centre for Global Chronic Conditions, London School of Hygiene & Tropical Medicine, London, United Kingdom

* dr.lavanya.vijayasingham149@gmail.com

**Data Availability Statement:** All data are in the manuscript and/or supporting information files.

**Funding:** The authors received no specific funding for this work.

## Abstract

Humanitarian health actors are beginning to better consider and manage non-communicable diseases, such as diabetes, in emergency and protracted crisis settings. However, a focus on the more globally prevalent type 2 diabetes (T2D) dominates. Blind spots prevail in the unmet needs for type 1 diabetes (T1D), a chronic autoimmune condition where individuals are unable to produce insulin, thereby dependent on lifelong insulin therapy and blood glucose management. Although some T1D management requirements overlap with those of T2D, the immediate risk of fatal complications following insulin therapy disruption, the earlier age of onset during childhood, adolescence or young adulthood, and its lower prevalence compared to T2D within communities and local health systems mean that T1D requires nuanced consideration and targeted interventions. Intending to inform program and policy design for people with T1D (PWT1D), we synthesized themes of lived experience from PLWT1D and their caregivers, and the tacit working knowledge of health providers and policymakers in the context of local humanitarian operations. Through a strategic search of health databases (up to July 2023), we identified 11 articles that include interview excerpts from PWT1D, caregivers, healthcare providers and policymakers about T1D management in humanitarian settings. We used reflexive thematic analysis to guide data extraction, coding, and synthesis, resulting in the identification of four overarching themes: food and insulin security, family relations, knowledge translation, and response to diagnosis. The narratives highlight harsh trade-offs made by PWT1D and their families in the face of insulin and food insecurity, as well as the damaging impact of low T1D education in families, communities and health systems. Targeted family and community-based solutions are urgently required, alongside systemic reforms and international collaboration to enable better T1D coping and management in humanitarian settings.

**Competing interests:** The authors have declared
that no competing interests exists.

## Introduction

In 2022, about 275 million people globally were estimated to need assistance as a result of a humanitarian crisis–increasing from 235 million in 2021 [1]. A humanitarian crisis or emergency refers to a singular or a series of events in a country or region, that causes serious disruption to the functioning of a society, resulting in human, material, or environmental losses which exceed the ability of affected people to cope using their own resources [2]. These crises are diverse, and can be categorised by its speed of onset (sudden or slow), its length (protracted) or cause (natural or man-made hazard or armed conflict, [2, 3]. Ultimately, these dynamics tend to present a critical threat to health and well-being of the affected populations.

Humanitarian health operations have only recently begun to better manage non-communicable diseases such as heart disease and diabetes–a group of conditions that has been mostly 'forgotten' in past humanitarian operations [4–6]. The burden of diabetes morbidity and mortality is high–in over half of humanitarian settings, diabetes prevalence is more than 10%, and in some international non-governmental organization (NGO) supported physical rehabilitation centres, diabetes causes more than 25% of lower limb amputations [7]. Indeed, humanitarian crises and settings whether acute or protracted, are typically characterised by complex, disrupted and fast-changing dynamics, that physically, logistically, financially, and politically impede governments and other non-state actors from providing sufficient and continuous care to the population. National health systems often have limited local government funding, and more recently, receive lower levels of international donor funding [8, 9]. The local health workforce tends to be under-capacitated and face high turnovers [9]. Logistical and supply chain limitations affect the availability of basic diagnostic equipment such as blood glucose monitors and life-saving therapies such as insulin [5]. The populations requiring care for NCDs often encounter significant financial obstacles when accessing private healthcare services. These challenges include out-of-pocket expenses for medical treatment and additional indirect costs such as transportation. Furthermore, individuals may also experience a loss of income due to the need to take time off work when seeking care-particularly in the context of daily wage employment, where they may be deterred from health-seeking even through public or subsidized healthcare delivery programs or schemes. [9–12].

Amidst these contextual barriers local health systems and humanitarian operations tend to place a stronger focus on the more prevalent type of diabetes—type 2 diabetes (T2D). Type 1 diabetes (T1D), also sometimes known as juvenile diabetes or insulin-dependent diabetes mellitus, is a chronic autoimmune disease. Nearly 9 million people are estimated to live with T1D globally- but T1D is also known to be grossly underdiagnosed and understudied in many world regions due to strained health and research infrastructure [13, 14]. The main difference between T1D and T2D is that in the former, the body has no ability to produce insulin, and in the latter, the body has a reduced ability to produce sufficient insulin. In T1D, the beta cells in the pancreas are destroyed by an autoimmune assault, often thought to be a consequence of viral infection, environmental toxins, or genetic predisposition [15]. Destruction of beta cells results in the body's inability to produce insulin, making the body incapable of blood glucose regulation, leading to glycaemic variability (GV) [16]. In T2D, the pancreas either does not produce enough insulin, or the body has become resistant to insulin [17]. Insulin resistance in T2D also leads to GV, but in milder or less advanced stages, T2D can present with less volatile and less life-threatening GV, compared to that in T1D [18]. Thus, while some people living with T2D may be able to manage their condition with oral medication and health behaviour changes, the only way to manage T1D is through the use of insulin therapy [19, 20]. Without insulin, T1D always leads to death (Beran et al., 2016; Willner et al., 2020). Furthermore, unlike T2D, there is no known way to prevent T1D. This is based on current understandings of T1D

aetiology which includes a complex combination of viral and environmental triggers, as well as phenotype and genetic differences that influence individual risk [13].

Due to the nature of the condition outlined above, all PWT1D require insulin to survive. Insulin deprivation is the most prevalent cause of death for children with T1D in low-income countries (LIC) [21]. In 2021, the average life expectancy of a ten-year-old diagnosed with T1D was 13 years in LICs, compared to 65 years in high-income countries [14]. While there are some variations in presentation and age of onset across different geographical regions, in low-and middle-income countries (LMIC), where most humanitarian operations reside, there are approximately 1.2 million people with T1D (PWT1D), with more than 50% under the age of 15 years old [13, 22, 23]. In humanitarian settings, where access to high quality and timely healthcare, diagnostic resources, and medication, is often fragmented and unaffordable, PLWT1D face many barriers to living long, productive, and disability-free lives.

Additionally, researchers have also hypothesized that patients with lower or non-existent insulin secretory capacity (e.g., T1D) are more sensitive to stressful environments (including emergency events in humanitarian settings and circumstances) than those with some insulin production capacity (e.g., T2D) [24]. Distressing life events, including injury and health-related events (e.g. severe accidents, death of a family member), are also known to be associated with an increased risk of developing T1D [25]. Population-based prospective and retrospective studies have shown that stressful life events (e.g. war) during the first 14 years of life may pose a particular risk for developing T1D [26, 27]. After the 2004 Marmara Earthquake in Turkey, the average HbA1c of 88 PWT1D increased from 7.4 pre-earthquake to 8.5 three months after the earthquake [28]. Twelve months before and after the 2007 flooding of Hull in the United Kingdom, 60 PWT1D experienced an average HbA1c increase of 8.1 to 8.6 [29]. Similarly, after the 2011 Great East Japan Earthquake, the average HbA1c of 55 PWT1D rose from 7.8 pre-earthquake to 8.1 three months after the earthquake [24]. Notably, those with T2D did not experience any difference in HbA1c after the Great East Japan Earthquake (7.3 vs. 7.3), which could imply that PWT1D in disaster settings have increased glycaemic vulnerability compared to people with T2D.

However, despite these differences in need and risk, there is still a paucity of research and guidelines that serve PWT1D in humanitarian settings [30]. In the World Health Organisation's (WHO) Package of Essential Noncommunicable Disease Interventions for Primary Health Care (WHO PEN), there is only one condition-specific recommendation for T1D, "Self-monitoring and self-adjustment of dosage is recommended in type 1 diabetes according to an agreed action plan with a health professional" (p. 57) [31] where service delivery may be compromised in an emergency or humanitarian setting. None of the targets, indicators, or policy options provided in the WHO GAP are specifically oriented towards T1D management. More encouragingly in humanitarian, emergency and resource-limited settings, a package of essential NCD interventions for humanitarian settings (PEN-H), developed by the International Rescue Committee (IRC) and the USAID, outlines the clinical management of T1D [32]. Similarly, the UN United Nations High Commissioner for Refugees (The UN Refugee Agency-UNHCR), IRC, and Informal Inter-Agency Group on NCDs in Humanitarian Settings outlines PWT1D as those at 'immediate risk' if there is interruption to care or when insulin access is compromised [33].

These guidelines and references focus on the clinical management, but do not include many discussions or consideration of lived experience of illness management or coping in people's local context. The lived experience of PWT1D, and their caregivers, as well as the tacit work knowledge of medical professionals and policymakers can inform the development of more comprehensive guidelines, programs and policies aimed at promoting health and reducing health complications of PWT1D in emergency and humanitarian settings. Based on our

assessment of literature, most studies about the lived experiences of PWT1D have either been conducted in high-income countries, or in stable settings within low- and-middle-income countries. Evidence about T1D in humanitarian settings is in its infancy, and tends to focus on the biomedical, quantifiable aspect of diabetes [30]. Little research addresses how T1D manifests or progresses during crises. Even less research examines the unique challenges of T1D management in crisis-affected settings where PWT1D have little control over their diet and access to essential medications and supplies. While a modest number of papers have assessed HbA1c changes and medication adherence in people with diabetes in humanitarian crises, to the best of our knowledge, it appears that no reviews have been published on the lived experiences of PWT1D to date.

This paper is a review and synthesis of the unique experiences of people living with and managing T1D amidst the contextual realities of humanitarian crises, with an aim to support policy development and program implementation. While highlighting the lived experiences of individuals and caregivers, we also include perspectives from healthcare providers and policymakers to provide a broader view of the structural factors that influence health-seeking behaviours and engagement with health resources.

## Methods

We conducted a qualitative synthesis of lived experience of PWT1D and their caregivers, as well as the tacit working knowledge of managing T1D in humanitarian settings. Using a reflexive thematic analysis (RTA) [34], we viewed and analysed the data from a constructivist paradigm, with a subjectivist epistemology, a relativist ontology, and an interpretive methodology [35]. The reflexive approach features our active roles, as researchers, in the knowledge production. RTA is considered a reflection of researchers' interpretive analysis at the intersection of 1) the data; 2) the theoretical assumptions underlying the analysis; and 3) the resources and skills of the researchers [34]. As such, in this approach, our subjective experience of living with T1D (author 1) and other chronic illnesses (author 3), as well as working on humanitarian programs and research on NCDs(authors 2 & 3) guide our engagement with the retrieved narratives and themes. We detail our positionality and reflexivity statement in a sub section below.

### Conceptual background

Narratives of lived experience of illness, caregiving, and tacit knowledge of health care stakeholders within the social context

For individuals living with a chronic illness, management and coping abilities are context dependent. Experiences evolve over time and life course, with narratives that often contain themes of disruption, diminishment, and discontinuous coping, due to a lack of access to coping resources [36–39] Illness caregiving, especially amidst gaps in health and social care, is largely unpaid and informal, mostly delivered by family and friends, and often viewed as a moral choice or imperative [40, 41].

The tacit knowledge from medical professionals, social care providers, and policy makers reveal the embedded, every-day knowledge acquired from working within a particular context, making the information explicit and known [42, 43]. Overall, an amalgamation of personal biomedical, popular, socio-cultural, religious, moral, economic, and political influences shape the meanings and value judgements attributed to illness, and consequently the responsive action at individual, population, and systems levels [44–46].

T1D care challenges in humanitarian crises: a social determinants and health systems building blocks perspective

As outlined by the WHO, 'health is determined by the conditions in which people are born, grow, work, live, and age, and the wider set of forces and systems shaping the conditions of daily life- the social determinants of health (SDoH) [47]. Under the Healthy People 2030 framework, SDoH are categorized into five different components: 1) Education access and quality; 2) Economic Stability; 3) Neighbourhood and Built Environment; 4) Social and community context; and 5) Health care access and quality [48]. Considering the aetiology of T1D, and applying the SDoH framework as a analytic lens, we explore how individual ability to manage and cope with T1D (e.g., administering insulin and adhering to a healthy diet) is influenced by a variety of ecological factors such as social and community contexts (e.g., gender norms), economic stability (e.g., poverty) and healthcare access and quality (e.g., access to providers trained in T1D care).

From a health access perspective, effective management of T1D is contingent on continuity of care within the local health system and ecosystem [49]. The WHO's six building blocks of health systems framework was used as a second analytical framework to help guide the synthesis of qualitative data [50, 51]. The framework describes the health system in terms of the six following components: 1) Service delivery; 2) Health workforce; 3) Health information systems; 4) Access to essential medicines; 5) Financing; and 6) Leadership and governance. Together, the SDoH and the WHO's six building blocks of health systems frameworks helped guide the analysis process.

Deep engagement and critical reflections on the multiple and accumulative influences of the social and lived context is necessary to fully understand how and why illness produces different sets of consequences. Cumulatively, economic, political, and climate instability within humanitarian settings impact all domains of T1D care. First, health infrastructure such as prescription, pharmaceutical and delivery services may be disrupted, resulting in the rationing of insulin and other diabetes supplies (e.g., blood glucose meters, insulin needles) [52]. Rationing of insulin and glucose test supplies has also been reported in several countries with more stable health systems, where health financing and access schemes do not ensure an affordable, adequate or continuous supply (USA, Panama, India, and Canada) [53]. Disruption of food and potable water is also a significant danger to PWT1D, who are at risk of life-threatening GV without consistent, balanced meals [54]. Climate disasters, political instability, and military conflict also often result in the reallocation of people and resources, leading to overburdened or abandoned health systems impacting the availability and quality of diabetes care [55].

## Data sources & search strategy

To identify studies reporting on the views and experiences of people with T1D, a search strategy for qualitative and mixed-methods articles was developed. The electronic search was run in five databases: Medline, Embase, Web of Science, Scopus, and PsychINFO. Databases were searched from their inception to July 2023, and search terms were related to T1D and humanitarian settings. After considerable consultation with a (masked) university librarian, a search strategy was crafted for Medline and modified for other databases [Supplementary Information 1].

Inclusion criteria for publications included:

1. Published in peer-reviewed journals;

2. Containing qualitative data and interview quotes;

3. In English with no year restrictions;

4. That reported on the lived experiences of PWT1D, their family members, caregivers, or healthcare providers; and

5. Were based in a humanitarian setting impacted by international or domestic conflict, economic or political instability, infectious disease outbreaks, refugee crises or climate disasters. Articles that included qualitative data from people with T1D and T2D, or did not differentiate between T1D and T2D, were also included and evaluated.

## Researcher reflexivity

Reflexivity refers to how the researcher and research process can shape the data collection, evidence synthesis, and conclusions drawn [56]. In line with the constructionist paradigm used that emphasizes the role of researcher subjectivity, we offer some reflections on how our own personal and professional histories inform our subjective interpretation and analytic lens.

(Author 1) is the primary researcher in this article, which was conducted as a part of her MSc in Public Health at (masked). The following is her reflexivity statement:

> I am a young, Canadian woman who has lived with T1D since the age of ten. I have always had access to insulin and diabetes supplies through publicly funded healthcare systems in Canada and the United Kingdom. My lived experience with T1D has been one of privilege; I have never experienced food insecurity, had to ration my insulin, or been personally impacted by humanitarian crises. I have also been the grateful recipient of extensive diabetes management training, both through hospital programs as a patient, and of my own volition as a diabetes researcher. For many years, I have shared my experiences living with T1D on social media. I cherish the online diabetes community—they have provided me with non-judgemental support and insight that has transcended languages and health systems. Social media has opened my eyes to the many, diverse, experiences of PWT1D accepting their diagnosis, navigating health systems, accessing insulin and medical technology, and integrating diabetes management into their lives. Many of their experiences contrast my own experiences.

> On one hand, I am an insider, intimately aware of the physical and mental burdens of living with and managing T1D. On the other hand, I am an outsider, having only experienced T1D in settings characterized by stability and access. I am mindful that the lived experiences of PWT1D in humanitarian settings are very different from my circumstances–I read their interview excerpts with humility and recognition of my position. My research lens thus stems both from my personal experiences as a PWT1D and my professional experiences as a public health researcher.

Author 2 works for a humanitarian organization and is involved in humanitarian NCD programming in Lebanon. Author 3 supervised the MSc Public Health research project, providing insights from ongoing research on NCDs in humanitarian settings and her prior work on chronic illness lived experiences, which similarly drew on her personal experience of living with and researching another chronic autoimmune illness in a middle-income country.

**Data extraction and synthesis.** We extracted the following material from all the included studies: year of publication, summary of the study's aim, country of study, study context, methods of evaluation (e.g., semi-structured interviews), study population (e.g., PWT1D, family member) [Table 1], and quotes about participants' experience of T1D in humanitarian settings.

We found that a reflexive thematic analysis (RTA) [34] was well-suited to achieve our research aims: to highlight the lived experiences of PWT1D and their caregivers, and the tacit

**Table 1. Information and characteristics of included studies.**

| Author | Title | Study Aim | Country/Setting | Qualitative Methods | Study population |
|---|---|---|---|---|---|
| [59] | Experience of living with type 1 diabetes in a low-income country: a qualitative study from Liberia | To better understand the psychosocial and economic impact of living with T1D in Liberia. | Liberia; Aftermath of the first Liberian Civil War | Qualitative study: semi-structured interviews | PWT1D: n = 10 Caregiver: n = 5 Provider: n = 10 Policy-maker: n = 1 Civil society organizations: n = 2 |
| [60] | Health system challenges for the management of cardiovascular disease and diabetes: an empirical qualitative study from Syria | To explore health system barriers to effective management of cardiovascular disease and diabetes in Syria. | Syria; Syrian Civil War | Qualitative study: document review; semi-structured key informant interviews; fieldwork in clinics | DM: n = 24 Providers: n = 12 *Distinction between T1D and T2D not provided |
| [61] | "Wasting away": Diabetes, food insecurity, and medical insecurity in the Somali Region of Ethiopia | To investigate rising concerns about diabetes among Somalis in eastern Ethiopia. | Ethiopia; Ongoing conflict in Ethiopia combined with drought and flood events | Qualitative study: ethnographic semi-structured interviews; participant observation; informal conversations | DM: n = 16 Siblings: n = 3 Providers: n = 4 Policy-makers: n = 5 Hospital Administrator: n = 1 *Distinction between T1D and T2D not provided |
| [62] | Societal Norms and Conditions and Their Influence on Daily Life in Children with Type 1 Diabetes in the West Bank in Palestine | To explore the experiences of daily life in children with type 1 diabetes (T1D) and their parents living in the West Bank in Palestine. | Palestine; West Bank in Palestine | Qualitative study: semi-structured interviews | PWT1D: n = 10 Parents: n = 10 |
| [63] | "I try the one that they say is good."—factors influencing the choice of health care provider and pathways to diabetes care for Syrian refugees in Lebanon | To explore factors influencing the choice of and pathways to diabetes care amongst Syrian diabetes patients visiting MSF clinics in Lebanon. | Syria; Syrian refugees in Lebanon | Qualitative study: In-depth interviews | PWT1D: n = 11 (T2D: n = 18) |
| [64] | "My heart burns"–A qualitative study of perceptions and experiences of type 1 diabetes among children and youths in Tajikistan | To explore and describe perceptions and experiences of living with type 1 Diabetes Mellitus among children and youths in Tajikistan. | Tajikistan; Aftermath of the 1997 Civil War | Qualitative study: semi-structured interviews | PWT1D: n = 18 Parents: n = 19 Provider: n = 4 |
| [65] | Self-reported medication adherence among patients with diabetes or hypertension, Médecins Sans Frontières Shatila refugee camp, Beirut, Lebanon: A mixed-methods study | To determine self-reported medication adherence prevalence and its predictors among patients with diabetes or hypertension in the Shatila refugee camp in Beirut, Lebanon. | Lebanon; Shatila refugee camp in Beirut | Mixed methods study: in-depth interviews | DM: n = 14 *Distinction between T1D and T2D not provided |
| [66] | Diabetes care in a complex humanitarian emergency setting: a qualitative evaluation | To explore patient and provider perspectives on diabetes health care implemented by MSF in the Democratic Republic of Congo. | Democratic Republic of Congo (DRC); Ongoing conflict between government and rebel forces | Mixed methods study: Focus group discussions; individual semi-structured qualitative interviews | DM: n = 7 Providers: n = 10 *Distinction between T1D and T2D not provided |
| [67] | Thoughts and attitudes toward disasters among Japanese patients with type 1 diabetes: A qualitative descriptive study | To explore the thoughts and attitudes of Japanese patients with type 1 diabetes during disasters. | Japan; Exploration of Hypothetical T1D management during a crisis | Qualitative descriptive study: semi-structured interviews | PWT1D: n = 10 |
| [68] | Type 1 diabetes patient experiences and management practices during the COVID-19 pandemic in rural Uganda | To investigate diabetes management of PWT1D during the COVID-19 pandemic lockdown in a rural district of southwestern Uganda. | Uganda; Access to and provision of international aid during COVID-19 pandemic | Qualitative study: semi-structured interviews | PWT1D: n = 3 Caregivers: n = 3 Providers: n = 2 |

*(Continued)*

**Table 1.** (Continued)

| Author | Title | Study Aim | Country/ Setting | Qualitative Methods | Study population |
|---|---|---|---|---|---|
| [69] | Diabetes management in the face of adversity: Experiences of asylum-seekers in Belgian reception centres | To examine how displaced people with diabetes experienced managing their illness before and throughout the process of fleeing their home communities. | Belgium; Asylum-seekers prior to arriving at Belgian reception centres | Qualitative study: Open and closed interviews | DM: n = 20 *Distinction between T1D and T2D not provided* |

*PWT1D = Person/people with type 1 diabetes

*T2D = Type 2 diabetes

*DM = Diabetes mellitus (includes type 1 and type 2 diabetes)

knowledge of health stakeholders in humanitarian settings [57]. RTA is an interpretive approach, within a constructionist paradigm. In the RTA analysis process, our subjectivity as researchers is understood to be integral aspect of the analysis process. In RTA, themes are not deductive or predetermined. Instead, inductive code and themes are sought, and then organized around "central organization concepts" that are identified from the data [34]. The conceptual frameworks were only used to provide a broad gaze on social, contextual and health-system related factors that contribute to lived experiences.

We applied Braun and Clarke's 6-step RTA process in the analysis: 1) Become familiar with the data, 2) Generate initial codes, 3) Construct themes, 4) Review potential themes, 5) Define and name themes, and 6) Write-up [58]. After familiarization with the literature, qualitative data from all included studies was aggregated into one document, printed out, and cut into individual quotes [Supplementary Information 2]. Initial codes were generated on a line-by-line basis using a pen and paper. After the articles underwent the initial stage of coding, additional re-reads were conducted, and updated codes were collated into candidate themes. Through ongoing reflection, the code groupings were amended until a smaller set of themes and subthemes remained. After data saturation was reached, themes and subthemes were established.

## Results

### Search results

The database searches resulted in 912 publications, from which 184 duplicates were removed. The remaining 732 articles were then screened by abstracts, and 93 articles were selected for full-text review. References from these articles were reviewed, and four additional studies were added. After a more thorough review, 11 studies that met the inclusion criteria were identified and chosen for RTA. A flow chart of the literature search process is presented in Fig 1.

### Study characteristics

Of the 11 studies identified, six studies were conducted in Asia (55%), four in Africa (36%), and one in Europe (9%). Qualitative data collected from ten different countries: Syria, Tajikistan, Lebanon, Palestine, Japan, Ethiopia, the Democratic Republic of Congo, Liberia, Uganda, and Belgium. Study contexts included civil war (27%), ongoing conflict (27%), refugee and asylum settings (27%), COVID-19 (9%), and one hypothetical pre-disaster setting (9%). Two study designs were included, qualitative studies (8, 82%) and mixed-methods studies (2, 18%). Just under half the studies (5, 45%) exclusively included the views of PWT1D, while the others (6, 55%) contained views from both PWT1D and PWT2D, or included people with diabetes

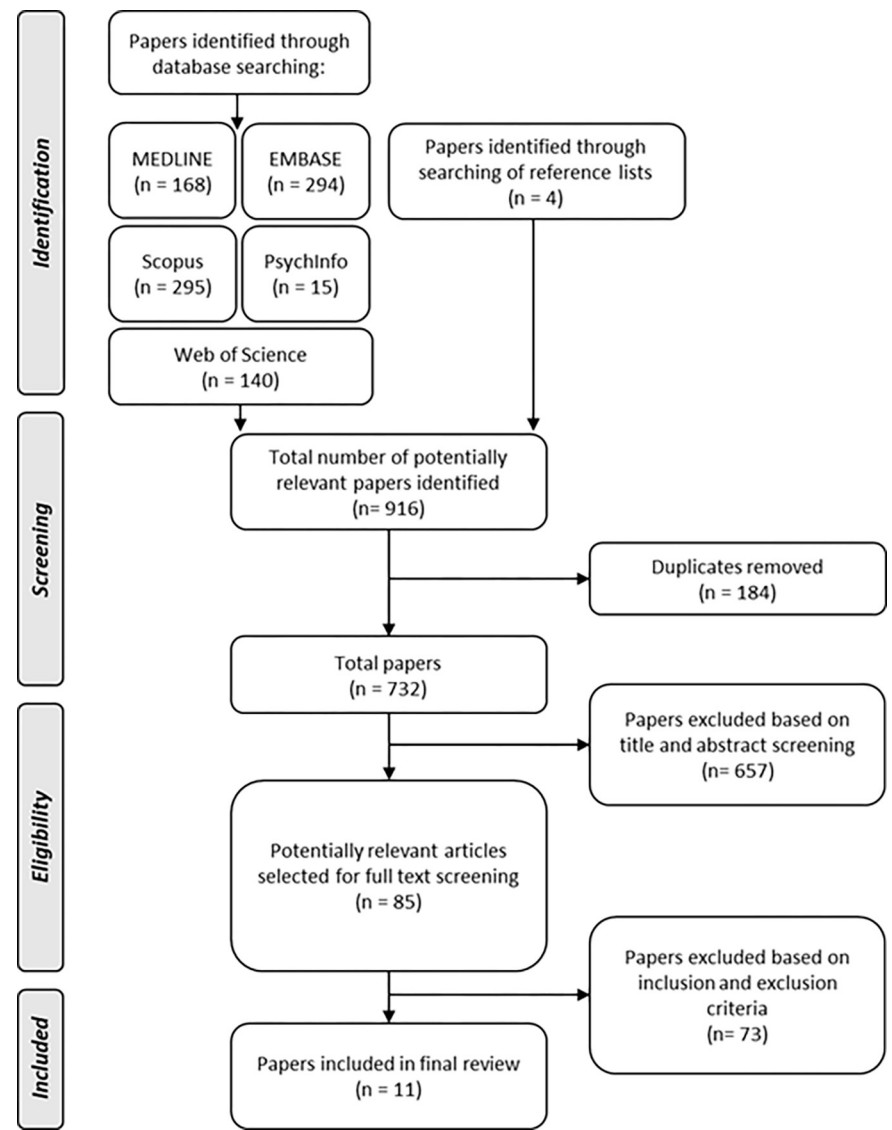

**Fig 1. Literature search strategy.**

mellitus (DM) with no distinction in type provided. All studies were published in 2014 or later.

## Quality appraisal

We used the consolidated criteria for reporting qualitative studies (COREQ) checklist [70] to assess the quality and transparency of the included studies. The COREQ Checklist is structured into three distinct domains, 1) Research team and reflexivity; 2) Study design; and 3) Analysis and findings. The assessment of these items aimed to provide insights into the potential biases and rigor of the included studies, without serving as exclusion criteria.

Domain 1: characteristics of the research team and their relationship with the study participants.

Most studies clearly indicated which researchers conducted the interviews or focus groups, and about half of them outlined the researcher's expertise by describing their experiences or

training. The studies provided moderate but adequate descriptions of the researchers' personal characteristics. Describing the relationship with the participants, however, was a low priority among nearly all the studies. Few studies mentioned when relationships with the participants were established, or whether participants were informed about the goals of the researchers. As such, it is not possible to evaluate the nature or dynamics of the researcher-participant interactions, and whether important factors such as power dynamics or conflicts of interest may have impacted data collection. Further, none of the studies contained reflexivity statements from the researchers. Transparency about how researchers' experiences and assumptions may have influenced the research process and findings was not prioritized.

Domain 2: study design and methodology

All studies described a clear methodological orientation, as well as a sampling strategy, sample size, data collection setting, and method of approach. However, one common gap among the included studies was information about non-participation. Few studies mentioned non-participation rates, and none of them explored the reasons for non-participation in detail. As a result, there is a reduced capacity to identify potential biases and assess the generalizability of the findings.

Domain 3: data analysis and findings.

Across all studies, description of the data analysis process and derivation of themes was clear. There were no identifiable gaps regarding the clarity and rigour of data coding. Notably, one commonality among the studies was the absence of information regarding participant feedback. Either this step was not deemed important to describe, was not able to occur, or did not occur. Participant feedback is an important step that improves credibility of the data and addresses potential biases. It also enriches the research findings by strengthening trust and rapport with the participants while validating their voices. The absence of this information could lead to a misrepresentation of participant experiences, potentially compromising the credibility and validity of study findings. Finally, reporting processes were strong. All studies presented consistency between the data and the findings, as well as clarity in major and minor themes.

## Synthesis of qualitative studies

Four distinct, but related, themes were identified from the thematic analysis: 1) Food and insulin security, 2) Family caregiving, 3) Knowledge translation, and 4) Response to diagnosis. Themes with illustrative quotes are presented in Fig 2.

## Theme 1: Food and Insulin Security

**Insulin.**   *"I could only come to pick supplies when [my parents] had money for me. There was a time I ran out of insulin even when I had tried to use it very sparingly, after realizing my parents had not got money for me. . ."*

*PWT1D [68]*

Insulin insecurity was one of the most prominent themes identified in this review. In several studies, patients reported rationing their insulin intake to make their supply last longer; [59, 68] a dangerous practice that can result in disability and death. Unfortunately, the phenomenon of '*rationing insulin*' is widely reported across LMICs where insulin can be unavailable, unaffordable, inaccessible, or rendered ineffective due to hermos-instability [52]. At times, insulin was so out of reach for patients that providers told PWT1D that they were likely going to die.

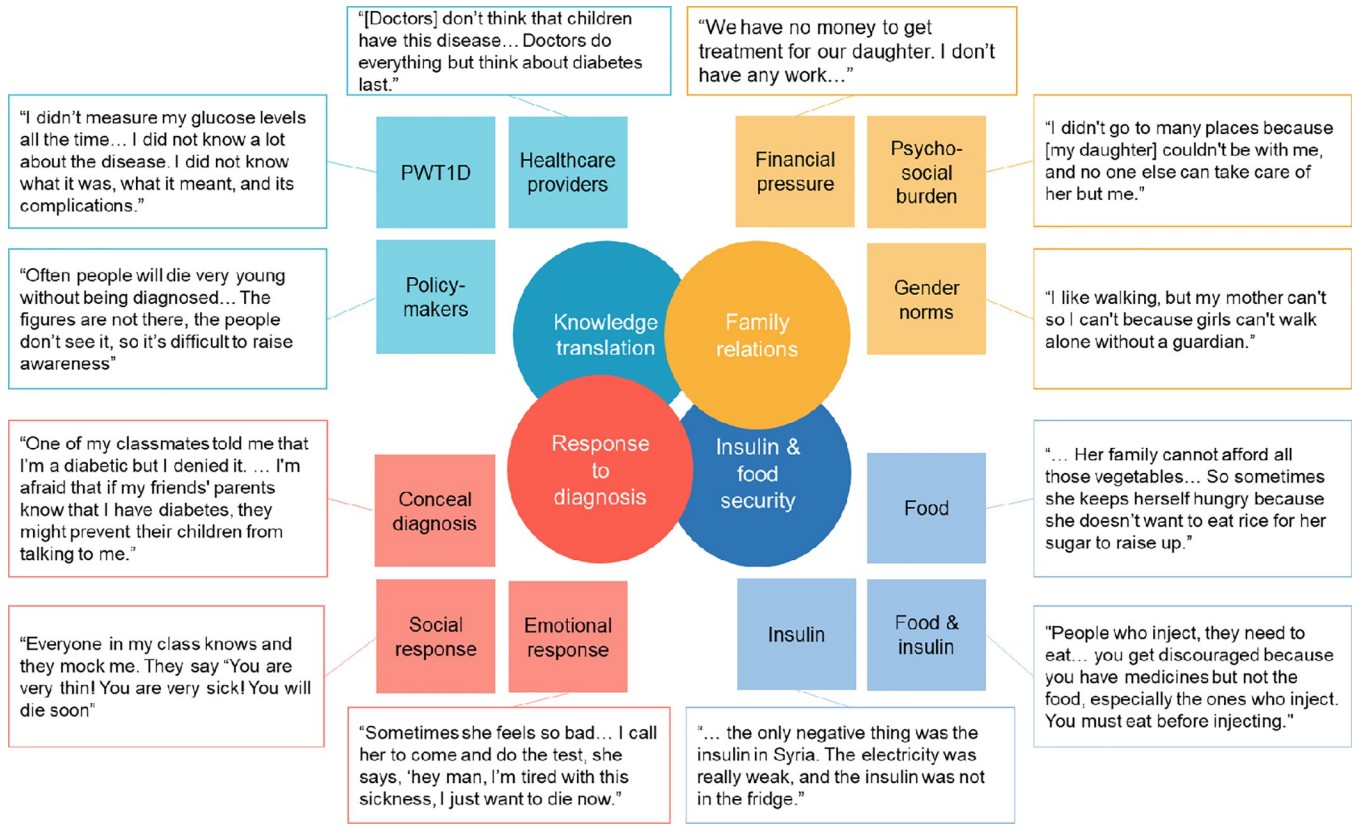

**Fig 2. Illustrative quotes from thematic analysis.**

*"If you have patients buying their own insulin, especially with the economy we are faced with in this country, it's going to be difficult telling most patients there is nothing we can do but just to allow you to die."*

*Healthcare provider [59]*

Even when PWT1D could afford insulin, transportation posed another significant barrier to access. During the COVID-19 lockdown in Uganda, vehicle restrictions meant that trips to replenish insulin had to be done by foot, or by expensive hired vehicles.

*"With the initial total lockdown, there were no means to travel but foot. . .. After easing some travel restrictions, the fares have doubled, and you must compete like never before."*

*PWT1D [68]*

Trips seeking health services could be dangerous. At times, they required payments to corrupt local enforcement officers.

*"You had to deal with LDU [COVID-19 guidelines enforcement]. You would have to plead with them or get them something for your passage [corrupt through your way through] . . .."*

*PWT1D [68]*

Importantly, even when insulin was available in humanitarian settings, its sensitivity to temperature could render it useless in situations of prolonged heat or cold.

> "... the only negative thing was the insulin in Syria. The electricity was really weak, and the insulin was not in the fridge."
>
> PWT1D [63]

Although emerging research has shown that insulin is more heat-stable than previously thought, cold storage still remains a barrier for many PWT1D in low resource settings [71–73]. In one article, only 30% of the PWT1D interviewed reported having access to a refrigerator for storage, and even then, spoke of problems with access to electricity [59]. PWT1D used to context- adapted alternative ways of keeping their insulin cool, such as storing their insulin coolers in banana trees by cutting small openings in the trunks [59]. Indeed, in the absence of refrigeration, adaptations such as the use of clay-pots and other evaporative cooling techeniques in low-resource settings have been found to be effective as a cooling strategy for insulin storage [72, 74].

**Food.** Interview transcripts highlighted the intimate relationship between access to insulin and access to food. The lived experiences of insulin insecurity could not be divorced from experiences of food insecurity.

> "People who inject, they need to eat; they need to have certainty of meals. When they get home they worry, thinking they may die due to lack of food... You get discouraged because you have medicines but not the food..."
>
> –PWT1D [66]

When PWT1D have access to insulin but experience food insecurity, GV is a major concern [75]. If insulin is taken without food, hypoglycaemia (low blood glucose levels) will occur [76], but if food is taken without insulin, hyperglycaemia (high blood glucose levels) will occur [77]. A consistent, balanced diet that includes fibre and protein is important for maintaining glycaemic control.

> "See, sometimes we would eat two times and other times have only porridge for the day [lunch or supper], and I do not know how to match her insulin to that .... That has been a very big puzzle"
>
> - Caregiver [68]

Insulin dosing that is not aligned with carbohydrate intake can be fatal. Even in HICs where PWT1D have uninterrupted access to food, severe hypoglycaemia accounts for up to 10% of deaths among youths [78]. The fear of hypoglycaemia and hyperglycaemia mean that at times, PWT1D in humanitarian settings have the impossible choice of choosing between starvation or eating the "wrong" foods.

> "But for her, she's from a poor family. Her family cannot afford all those vegetables, those extra things to balance her diet. So sometimes she keeps herself hungry because she doesn't want to eat rice for her sugar to raise."
>
> –Interpreter for PWT1D [59]

This means that *even* when insulin is available, affordable, or provided without cost to patients, and stored correctly, food insecurity can render insulin therapy ineffective, or worse, life-threatening.

### Theme 2: Family Caregiving

**Financial pressure.** T1D's substantial financial burden is felt by all members of the family, often forcing parents to make unthinkable decisions about which children to support. In many cases, access to food, insulin, and medical supplies–the prerequisites of T1D management–is contingent on the ability to pay.

One father explained that he had to neglect his child's T1D care to provide for the rest of his family.

> *"We have no money to get treatment for our daughter. I don't have any work. We are advised to go to the hospital twice a year but this year we didn't go. The last time we were at the hospital two years ago, it did cost 2000 Somoni (430 USD).*
>
> *Father of a child with T1D [64]*

Where parents were able to pay for diabetes management supplies, it often required finding additional income streams.

> *"My husband has two jobs: at night he works in the municipality and during the day he works in the ice cream factory. Sometimes we don't have money, so we have to borrow money from relatives".*
>
> *- Mother of a child with T1D [62]*

Or taking financial support away from other children.

> *"Each month, 600 shekels is deducted for my son's treatment, although I have a big family. I always wonder how I will meet their needs. If my son did not have diabetes, I could use this money for my daughter, who is going to university".*
>
> *Father of a child with T1D [62]*

Many families needed to make appalling choices under the principle of opportunity cost–the choice between diabetes care for one child or meeting the basic needs of the family is a deplorable scenario that is all too common [79].

**Gender norms.** Gender expectations influenced parents' expectations for their daughters' futures, leading to differential treatment between sons and daughters diagnosed with T1D. Gender norms were a significant upstream barrier to T1D management and could not be overcome with individual-level facilitators like discipline.

For daughters, parents expressed anxiety about marriage prospects, citing that T1D compromises their daughter's perceived suitability as wives and mothers.

> *"Her father was so sad, he felt that she will be a burden to him; how will she get married? I don't think she will because she has diabetes."*
>
> *Mother of a child with T1D [62]*

In some humanitarian settings, gendered cultural expectations made it difficult for female adolescents, but not males, to be physically active. Restricting daughters from leaving home and preventing them from participating in physical activity was common.

*"I like walking, but my mother can't so I can't because girls can't walk alone without a guardian. Also, she prevented me from going out walking with my friends."*

*PWT1D [62]*

In some communities, public physical activity is unacceptable for women [62]. Women with T1D who break social conventions to try and improve their glycaemic control could face significant familial and social implications.

Mental health and psychosocial burden- including for caregivers

Many family members discussed the overwhelming difficulty of caring for loved ones with T1D.

*"[My wife] gets discouraged. She wonders for how long this will continue. She figures I am already gone, living on medicines all my life. It brings her worry. . . The diet brings a lot of strife. Because she has to prepare her food with the children, then prepare mine. It brings many quarrels."*

*PWT1D [66]*

Some parents, particularly mothers, spoke about how their child's diagnosis with T1D led them to reduce social contact with their communities. The unequal social and emotional burden on mothers further highlights the gendered differences in T1D management. Because community events were often centred around "unhealthy" food, it was difficult for children with T1D to stick to their approved diet. Thus, parents chose to sacrifice their attendance at social events to promote their child's diet adherence and protect their families from judgement.

*"For today, and tomorrow I will not go to social events such as weddings. Even my neighbours are disappointed with me because I don't visit them. Even my mother, I don't visit her, especially when my son doesn't commit to the treatment."*

*- Mother of a child with T1D [62]*

Social and community support for families managing T1D was often poor due to insufficient knowledge and T1D stereotypes, leading to feelings of depression, anxiety, and isolation among PWT1D and their families.

Many parents also noted significant distress from seeing their children receive insulin injections. Although many parents knew about the necessity of insulin for T1D management, seeing their child in pain led to missed insulin doses.

*"My daughter cried when I gave her injections. She cried "Don't do that! Don't do that!" Because my heart burned, I could not do the insulin injections."*

*Mother of a child with T1D [64]*

*"For almost one month he did not get insulin injections, because we just didn't want to hurt him anymore, but then he got very, very sick."*

*Father of a child with T1D [64]*

Few family members were aware of T1D complications and alleviating the immediate felt pain of injections was therefore prioritized over preventing long-term complications.

### Theme3: Knowledge Translation

**T1D Knowledge by PWT1D.**  A lack of knowledge translation, known as the dynamic process, that includes the synthesis, exchange, and application of knowledge to improve health, was a recurring theme across all eleven studies. Minimal knowledge about diabetes complications wasn't an experience unique to family members–many PWT1D were never provided with an opportunity to learn about what T1D was and how mismanagement could lead to complications.

*"[My diabetes] wasn't stable when I was young. I told you, I didn't know about eating rice and white bread. All of this increased diabetes levels. I didn't measure my glucose levels all the time. I did not know what it was, what it meant, and its complications. I used to know that I had to take the medication, eat normally and cut out sweets only. I did not know that carbohydrates increased the glucose level."*

*–PWT1D [63]T1D Knowledge by Healthcare Providers*

Low T1D knowledge at the patient level was indicative of a knowledge translation gap between healthcare providers and PWT1D, but also of a system-wide insufficiency in T1D knowledge.

One father described his despair when his daughter with undiagnosed T1D was transferred from facility to facility, receiving treatments for a variety of conditions–malaria, measles, intestinal worms, allergies, and typhoid fever–until finally falling into a coma due to healthcare provider-induced severe hyperglycaemia.

*"Wherever we had been, the hospital, the diagnostic centre and other places, nobody found out what was going on [. . .] When we went to this other hospital, they said they needed to "wash and clean" her stomach and she got a glucose infusion. After that she went into a coma."*

*Father of a child with T1D [64]*

The lack of knowledge about T1D and delayed diagnoses meant that many PWT1D in humanitarian settings experienced complications requiring serious clinical intervention. Because health services were often run by ad hoc health centres and short-term humanitarian aid clinics, PWT1D experienced little standardization in treatment:

*"Specialists have studied in different hospitals around the world, and each doctor follows the methodology taught at that hospital, resulting in a variety of procedures and prescriptions. Only a few doctors prefer to have standardized procedures."*

*–Ministry of Health Official [60]*

The combination of siloed healthcare providers and under-resourced health services also invited a lack of accountability, leading to medical "shortcuts".

*"Too many cases of diabetic foot amputation in Syria. . . The problem is that surgeons taking care of the diabetic foot have poor skills in treating infections. Therefore, they go with the easier solution, which is amputation instead of treatment, especially as there is no accountability."*

*–Ministry of Health Official [60]*

In high-income settings where health services are comparatively more accessible than in many humanitarian settings, people with diabetes who undergo a major amputation tend to have a 5-year survival rate of less than 50% [80].

T1D Knowledge by Policymakers

The lack of health providers trained about T1D reflects a long-standing prioritization of infectious diseases such as HIV, malaria, and tuberculosis. Due to under-funded health systems and resource allocation towards communication diseases, there is a paucity of data demonstrating the T1D burden in many humanitarian settings.

*"Often people will die very young without being diagnosed. As a result, people don't see those cases, and when you want to do advocacy they will tell you we don't see those cases. But we know those cases occur, but before they are diagnosed they pass off and they are buried and then no one knows them. . . The figures are not there, the people don't see it, so it's difficult to raise awareness, and because we are not responding, people are not aware, so it's a kind of circle. . ."*

*Policymaker, [59]*

In LMICs, T1D registries are rare. When data on diabetes is collected, T1D and T2D are generally grouped together, further hiding the true prevalence of T1D [59].

## Theme 4: Response to Diagnosis

T1D is not just hidden at the health-system data collection level. It is also commonly hidden on an intrapersonal level. Many of the PWT1D in the studies identified concealed their diagnosis from their social and familial network due to fears of rejection and stigmatization. Although there was sometimes pressure to disclose their diagnosis, PWT1D went out of their way to keep their diagnosis hidden.

**Concealing diagnosis.** *"One of my classmates told me that I'm a diabetic but I denied it. . . . I'm afraid that if my friends' parents know that I have diabetes, they might prevent their children from talking to me, and say I am sick".*

*PWT1D [62]*

School-aged youths with T1D identified alienation from friends and classmates as a central contributor to their psychosocial stress [59]. They reported anxiety about their inability to eat the same foods and do the same activities as their peers. Indeed, when T1D diagnoses were revealed at school, children faced consequences.

*"The teacher stopped me from playing football because he is afraid I will have hypoglycemia or become dizzy. . . Sometimes I cry, because I love sport."*

*PWT1D [62]*

Children both anticipated and experienced stigma due to their diagnoses. In some cases, youths with T1D isolated themselves from their friends for self-protection.

*"My life changed because since I was diagnosed in 2014, I stopped going out with my friends. . . its always 'My man, that stuff I can't eat, I don't do that.' Someone else, 'Oh my man you're selfish.' But I really didn't want to tell him why the reason I was doing it. . . And it hurt me a lot."*

*PWT1D [59]*

Fears of stigma and discrimination about T1D were so strong that some asserted that they would not tell others about their diagnosis, even in the case of a hypothetical disaster.

*"I do not want to say that I am diabetic when a disaster strikes."*

*PWT1D [67]*

Like the psychosocial burden felt by families of children with T1D, poor support in social contexts led to poor mental health outcomes for PWT1D.

**Emotional response.**   Dealing with shame, stigma, and exclusion are aspects of diabetes distress (DD), a condition used to refer to the accumulation of daily stresses that individuals with diabetes experience while managing their diabetes [81]. DD captures the feelings of powerlessness, overwhelm, frustration, and anger that result from living with T1D [82]. Though there are no published meta-analyses about the prevalence of DD among PWT1D, it is estimated that between 20 and 40% of the T1D population experience elevated DD [81].

*"My daughter is often very sad about her situation. Sometimes she says that she would rather die."*

*PWT1D [64]*

*"Sometimes she feels so bad. You know, when she's at play and I call her to come and do the test, she says, 'Hey man, I'm tired with this sickness, I just want to die now. Every day, all my fingers hurting, I'm tired."*

*Caregiver [59]*

Although DD varies across populations of PWT1D, being female, being young, and having weak social or familial support systems (e.g. living with an unsupportive partner, having an uninformed family, perceiving a lack of help in one's social support network) are significantly associated with higher levels of DD [83, 84]. These characteristics were common across the PWT1D interviewed in the included studies.

## Recommendations

This synthesis of lived experiences of PWT1D and their caregivers, and the tacit knowledge of health stakeholders in humanitarian settings reveals several barriers to coping and care: access to insulin and food, familial and cultural constraints, information gaps, clinical management, and illness coping. We discuss how these findings can be translated into policies and programs that address the unmet need for material, psychological and health support among PWT1D in humanitarian settings.

### Remove the need for 'trade-offs' by addressing economic costs, and acting on the multi-level commercial determinants of food and insulin insecurity

Glycaemic control is often discussed in relation to "medication adherence", defined as the extent to which patients follow prescribed medication dosing regimens [85]. As was highlighted in interview excerpts, adhering to T1D management routines in many humanitarian settings was often effectively unattainable due to food insecurity, insulin and daily glusose testing accessibility,, among other challenges. Rationing insulin was a common coping mechanism in the face of medication and food insecurity, a finding that is aligned with existing literature on T1D management strategies in under-resourced settings [52]. When insulin was not available, or food options were not conducive to glycaemic control, PWT1D sometimes opted to skip meals entirely.

Adherence, whether to healthy eating or to insulin regimes, is a privilege, and for many in low resource and humanitarian settings it is not a choice or agentic decision, but a trade-off [86]. Non-adherence narratives can inadvertently allocate blame to the patients, taking attention away from the significant, contextual barriers to diabetes management. Researchers, healthcare providers, and policymakers must understand the economic context in which these trade-offs are made.

Existing guidelines for NCD management in humanitarian settings can be updated to include these nuanced considerations of T1D experience and program needs. An immediate way to remove some constraints can be through providing customised packages of self-care and nutritional resources (e.g., vouchers or provisions to access transport and food banks or regular provision of nutritious meals, infrastructure to store insulin, supplies of test strips and needles). These provisions are necessary to address the economic component of T1D management outside of health facilities–that is, where people live and work. The bigger and more time-consuming task lies at the national structural and international trade policy level–that is to enforce policies and programs that *directly* target the factors blocking sustainable and affordable access to insulin and healthy food–the commercial determinants of insulin and healthy food.

Economic constraints, and catastrophic out-of-pocket health payments is not unique to T1D management, and is also experienced by people living with T1D outside of humanitarian settings to various degrees [53]. These overarching discussions relate to the management of T2D and many other NCDs, mental health and neurological conditions. There is ongoing policy calls and advocacy, such as through past and upcoming dialogues on 'Sustainable Financing for NCDs and Mental Health'. Chronic conditions can impose significant economic development and productivity burden, and more effective financing strategies are needed to address the NCD investment gaps globally [87].

### Provide family and community-based mental health support and awareness tools for individual self-care, family-based caregiving, and managing stigma and psychosocial distress

This theme runs parallel with the expanding literature base that characterizes T1D as a "family affair" due to the impact of its management regime on all members of the household [88, 89]. A key point of difference between T1D and T2D is the life course stage at which the disease is often diagnosed or managed at–childhood, adolescence, and young adulthood. Involving and engaging caregivers in the diagnosis and management processes is imperative to achieve positive illness coping. The family is the main source of medical, emotional, and financial support for PWT1D –particularly when health and social welfare systems fail at providing necessary care. Knowledge of family dynamics is critical when considering approaches to promoting

T1D care in humanitarian settings. Interventions that exclusively target PWT1D, rather than PWT1D with their family and caretakers, can fall short by failing to address management barriers related to familial or cultural norms. Healthcare providers and community health workers or volunteers need to be aware of the stigma and cultural contexts that shape family involvement in T1D management.

PWT1D, caregivers, and healthcare providers discussed distress related to the T1D diagnosis and subsequent management. This theme echoes findings from previous studies which have shown that PWT1D and their caregivers sometimes actively hide a T1D diagnosis for fear of being characterized as intolerable burdens (e.g., chronically sick, a reminder of death), inferior marriage material (e.g., high-risk pregnancies, unable to provide financially for the household), and depressing people to be around (e.g., unable to enjoy certain foods and activities) [62, 90]. Stigma is also strongly related to psychological distress among people with all types of diabetes [91]. Even in stable settings, PWT1D have significantly higher rates of mental health conditions including depression, anxiety, and suicide compared to those without T1D [92]. Poor mental health is associated with poor diabetes management and fewer health-seeking behaviours, both of which are important in humanitarian settings where health support systems are diminished [93]. These findings reiterate literature emphasizing the importance of post-diagnosis treatment support for people with chronic diseases and their families in under-resourced settings [94, 95].

Future programs and policies should not just be *patient-centred*, but also, *family-centred*. A strong focus on families, their education and psychosocial support in managing T1D as a family and household unit is required to promote sustainable T1D management. Healthcare providers should endeavour to create personalized T1D treatment plans that intimately considers the familial and cultural norms, while also investing in community and structural level awareness programs that can alleviate the social stigma surrounding PWT1D.

For this, culturally congruent, innovative, and frugal programs that facilitate the delivery of diabetes education and psychosocial support should be considered, especially where on-site health staff capacity is limited. These include peer support groups, e-health programs, and talk-therapy delivered by trained lay health workers or community members. Program and policies must be developed with the goal of fostering community awareness and education on T1D, dispelling myths related to the condition. Visible involvement and mass communications by trusted and respected community or religious leaders, and other positive social influencers can be leveraged to disseminate accurate information and encourage healthy and care-seeking behaviours. Peer support programs that integrate approaches for daily management and social and emotional support have been used to promote the management and coping of other chronic conditions [96, 97]. Additionally, lay-delivered talk therapies for PWT1D and their families can also be beneficial, with similar programs conducted in humanitarian settings being seen as feasible to implement, and acceptable by service users [98].

T1D is prevalent in girls and boys in near equal proportions, but gender norms may create inequalities in coping and management. Programs that include the promotion of gender equality and address restrictive gender norms that affect illness coping and management, including those involving men and boys, can also be beneficial for promoting individual and community-level T1D management [99]. Drawing lessons and evidence from the HIV sector, it is also important to act on structural, educational, and economic domains that can accumulatively contribute to positive health outcomes. This can include abolishing school fees, extending compulsory education, cash transfers, microfinance and income-generating activities such as skill-building and vocational training, programs to address gender-based violence, through community-based participatory learning, peer and partner discussions [100, 101]. Programs must be context-relevant and acceptable among the community, and should include societal

and cultural themes of experience such as anxieties related to marriage prospects and societal reputation. Importantly, these programs must be informed and designed based on the meaningful involvement of the people and families that live with T1D themselves—which is overall still a less common practice in humanitarian settings and many LMICs [102].

### Invest in and provide T1D relevant capacity building, knowledge generation and translation to practice for healthcare workforce and policymakers

Few studies provided insights into the tacit knowledge of healthcare workers and policy makers, suggesting that working knowledge about T1D was not strong among healthcare providers and policymakers across the humanitarian settings. As literature from other low-income settings has highlighted, healthcare providers often lack opportunities to learn fundamental knowledge about T1D, leading to misdiagnoses, limited patient education, and poor patient T1D management skills [64].

In humanitarian settings, low professional knowledge and uncoordinated clinical practices are accompanied by limited data and disease surveillance. In 2018, the Humanitarian NCD Interagency Study in Emergencies and Disasters (UNITED) found that less than 50% of 83 surveyed humanitarian sites distinguished between T1D and T2D, and less than 30% reported on GV complications [103], further supporting the necessity of strengthened T1D knowledge translation in humanitarian settings.

To address these gaps, efforts to sensitize healthcare providers at primary, community, and tertiary health care levels, and strengthen knowledge translation at provider and political levels, is necessary. This can include capacity-building and training initiatives, and accessible educational materials (e.g., infographics) about T1D stereotypes, insulin dosing, and complications. Creating and disseminating T1D management curriculums for healthcare providers (e.g., nurses, doctors, and community health workers) and establishing guidelines to standardize systems of care is necessary to improve the coordination and accountability of T1D care in humanitarian settings. The NCDI (non-communicable disease and injuries) Poverty Network and the PEN-PLUS partnership is an example of a global initiative that is mobilising resources, building awareness, and developing local health workforce capacity to reduce the burden of complex chronic conditions like T1D [104–106].

### Limitations

This qualitative synthesis has taken a transparent, systematic approach to synthesizing qualitative evidence about the experiences of PWT1D in humanitarian settings; filling a significant gap in the research base. Nevertheless, we acknowledge several limitations in our approach.

Despite a comprehensive search of five databases, relevant studies may have been missed. First, our definition of humanitarian setting and subsequent database search strategy may have missed relevant studies. This is not an exhaustive systematic review, but rather a qualitative synthesis of papers that richly portray the nuanced lived experiences of PWT1D, their caregivers, healthcare providers, and policymakers in humanitarian settings. Diabetes is also a heterogeneous disease with large variations in beta cell dysfunction and insulin resistance, which can lead to conflation between T1D and T2D diagnoses [107]. In humanitarian settings, sites also often only collect aggregate data on the number of diabetes consultations, rather than the number of patients with T1D or T2D, which can lead to the generalization of all diabetes to T2D [103]. Both circumstances can lead to the misclassification of T1D as T2D, meaning that articles excluded due to a focus on T2D would have been missed in the research process.

Further, grey literature and unpublished reports were not included in this meta-synthesis, which may have further provided insight into the lived experiences of PWT1D in humanitarian settings. Limiting this exercise to studies published in English also means that key insights provided in other languages were not included. A future review could expand upon this work by searching the grey literature and by including non-English publications.

Despite these limitations, this article provides a rich synthesis exploring the contextual, cultural, gendered, and health system factors that impact T1D management in humanitarian settings. Future research on how to address barriers to T1D management in humanitarian settings would be enhanced by more comprehensive focus on the experiences of the humanitarian organizations, donors, public and private health actors, and local NGOs who work in the field.

## Conclusion

This paper consolidates narratives of lived experience and tacit knowledge from PWT1D, their caregivers, healthcare providers, and policy makers, illuminating the mounting health needs of people impacted by humanitarian crises. The narratives and themes in this paper demonstrate how T1D management intimately reflects, and is contingent on the broader familial, cultural, and economic systems in which it exists. Policymakers and healthcare providers should be aware of the severe trade-offs that PWT1D and their families make in the name of diabetes management. In particular, NCD programs that strive to manage T1D must also include arrangements fro sustainable and continuous access to crucial needs such as insulin, glucose testing kits and strips, as well as nutritional goods or food vouchers. Additionally, effective management strategies in a younger (sometimes pediatric) population, with interventions that engage and involve their parents as caregivers, primary health care service providers, and broader members of society including their school peers, teachers, and neighbours is necessary.

The recommendations and insights outlined in this report are also relevant to the management of other NCDs, including T2D in many resource-constrained settings. Existing guidelines and frameworks for action can be updated to include notes on the differences and similarities of needs in managing T1D, T2D, other NCDs, mental health and neurological conditions, particularly where negative outcomes are known to be amplified by experiences of deprivation and poverty. Targeted action through joint multi-sectoral investment can address some of the root constraints and barriers to care outlined in this paper. We sincerely hope that these insights will contribute to the development of stronger T1D aid and resources for PLWT1D, their families, their communities, and health stakeholders in all forms of humanitarian settings.

## Supporting information

**S1 Text. Literature search strategy for medline (OVID) database.**
(DOCX)

**S2 Text. Images of manual analysis process-establishing codes, sub-themes, and themes.**
(DOCX)

## Author Contributions

**Conceptualization:** Lavanya Vijayasingham.

**Data curation:** Oria James.

**Formal analysis:** Oria James.

**Methodology:** Lavanya Vijayasingham.

**Supervision:** Lavanya Vijayasingham.

**Writing – original draft:** Oria James, Lavanya Vijayasingham.

**Writing – review & editing:** Linda Abbou-Abbas, Lavanya Vijayasingham.

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
