## [Decision Letter · Decision Letter 0]

17 Apr 2024

PGPH-D-24-00382

Living with and managing type 1 diabetes in humanitarian settings: a qualitative synthesis of lived experience and stakeholder tacit knowledge

Dear Dr. Vijayasingham,

Thank you for submitting your manuscript to PLOS Global Public Health. After careful consideration, we feel that it has merit but does not fully meet PLOS Global Public Health’s publication criteria as it currently stands. Therefore, we invite you to submit a revised version of the manuscript that addresses the points raised during the review process.

The manuscript has been evaluated by two reviewers, and their comments are available below.

Although both reviewers are positive about your manuscript, they have several requests for modifications. Please see the comments below and in the attached documents.

Could you please carefully revise the manuscript to address all comments raised?

Reviewer 1 notes that the discussion section consists largely of recommendations and suggests that the section could "either be left as it is, or it could be split into a discussion section (reflecting on the implications of the findings for practice), and a recommendations section (framed around the three recommendations that are presented)." Like reviewer 1, I do not have a strong opinion, and so leave the suggestion to the authors for consideration.

We look forward to receiving your revised manuscript.

Kind regards,

Steve Zimmerman, PhD

PLOS Staff Editor

Journal Requirements:

1. We noticed you have some minor occurrence of overlapping text with the following previous publication(s), which needs to be addressed:

-https://docs.google.com/document/d/1X-oA80EUZrE5SagOizqctp43KUrpr3CMRkWjkkl1RJo/edit

In your revision ensure you cite all your sources (including your own works), and quote or rephrase any duplicated text outside the methods section. Further consideration is dependent on these concerns being addressed.

Additional Editor Comments (if provided):

Reviewers' comments:

Reviewer's Responses to Questions

**Comments to the Author**

1. Does this manuscript meet PLOS Global Public Health’s publication criteria? Is the manuscript technically sound, and do the data support the conclusions? The manuscript must describe methodologically and ethically rigorous research with conclusions that are appropriately drawn based on the data presented.

Reviewer #1: Yes

Reviewer #2: Yes

2. Has the statistical analysis been performed appropriately and rigorously?

Reviewer #1: N/A

Reviewer #2: N/A

3. Have the authors made all data underlying the findings in their manuscript fully available (please refer to the Data Availability Statement at the start of the manuscript PDF file)?

Reviewer #1: Yes

Reviewer #2: Yes

4. Is the manuscript presented in an intelligible fashion and written in standard English?

Reviewer #1: Yes

Reviewer #2: Yes

5. Review Comments to the Author

Reviewer #1: This is an interesting and relevant manuscript; the topic is of increasing importance in humanitarian healthcare, especially since recent emergencies (Ukraine, Gaza) have highlighted the challenges associated with Insulin provision during the first 30 days of the emergency response. Bringing together perspectives of people with lived experience enriches the policy discussions in the build up to the UN high level meeting on NCDs in 2025.

The manuscript is well written, the analytic frame is justfied, and the final categorisation of themes makes sense. The findings are presented clearly and I believe will resonate well with other practitioners working in this field.

I have made a number of minor comments in the manuscript itself. My only substantial comment is about the discussion. The discussion as presented is more accurately described as a set of policy recommendations, supported by the findings of the study and by a broader literature on this subject. It reads well, and personally speaking I find it a helpful and powerful way to conclude the manuscript. But from a teminological point of view, it would be more accurately referred to as 'Recommendations' rather than 'Discussion'. In the end, it could either be left as it is, or it could be split into a discussion section (reflecting on the implications of the findings for practice), and a recommendations section (framed around the three recommendations that are presented). I do not have a strong opinion on this, I would suggest this is an editorial decision.

Reviewer #2: April 14, 2024

Dear Editors,

Thank you for the opportunity to review the article “Living with and managing type 1 diabetes in humanitarian settings: a qualitative synthesis of lived experience and stakeholder tacit knowledge.” This is very well written article that uses qualitative synthesis methods to provide a review of the challenges faced by persons with type 1 diabetes in humanitarian settings. While I have minor suggestions for considerations, I believe this work provides an important contribution to the literature highlighting the special considerations that are needed for persons living with type 1 diabetes. The most important point to consider is adding information on how the results of this synthesis compare to what is known about the management of type 2 diabetes in humanitarian settings. A strong case is made as to why it is important to consider type 1 diabetes separately but does not close the loop by describing how the findings differ from what is known for type 2 diabetes in humanitarian settings. Highlighting the differences between type 1 and type 2 diabetes in this context will emphasize the importance of this work.

Abstract:

The last sentence should be adjusted to better reflect the findings of the article. The terms “structural and international levels” are vague and difficult to understand and relate to what is presented in the preceding abstract paragraph.

Introduction:

Page 3, paragraph 2, last sentence. “Forgone daily wage associated with…” is not clear and authors should clarify what they are trying to get at – the need to seek care means they lose a day of work?

Page 3, Paragraph 3, first sentence. “Barriers and challenges” is redundant.

Page 4, Paragraph 1. I would be careful to over-generalize. There are persons with type 2 diabetes who without insulin for several days will similarly become extremely sick and become at risk for hyerosmolar coma.

Page 5 paragraph 1, first sentence. The word “manifestation” is not correct as it implies incidence. Whereas the following sentence discusses diabetes control (with level of hemoglobin A1c).

Page 6 paragraph 2. First sentence. Missing word “people” before living.

Methods and Results are well written, and I especially liked Figure 2 as a depiction of themes, subthemes and illustrative quotes.

Discussion

Also well written with clear delineation of recommendations. As noted previously my one strong recommendation would be to specifically identify how this differs from review of challenges and recommendations for type 2 diabetes.

6. PLOS authors have the option to publish the peer review history of their article (what does this mean?). If published, this will include your full peer review and any attached files.

**Do you want your identity to be public for this peer review?** For information about this choice, including consent withdrawal, please see our Privacy Policy.

Reviewer #1: No

Reviewer #2: No

---

## [Editor Report · Decision Letter 1]

30 May 2024

Living with and managing type 1 diabetes in humanitarian settings: a qualitative synthesis of lived experience and stakeholder tacit knowledge

PGPH-D-24-00382R1

Dear Dr. Vijayasingham,

We are pleased to inform you that your manuscript 'Living with and managing type 1 diabetes in humanitarian settings: a qualitative synthesis of lived experience and stakeholder tacit knowledge' has been provisionally accepted for publication in PLOS Global Public Health.

Best regards,

Julia Robinson

Staff Editor